# Studies on Three-Dimensional (3D) Accuracy Optimization and Repeatability of UAV in Complex Pit-Rim Landforms As Assisted by Oblique Imaging and RTK Positioning

**DOI:** 10.3390/s21238109

**Published:** 2021-12-04

**Authors:** Rui Bi, Shu Gan, Xiping Yuan, Raobo Li, Sha Gao, Weidong Luo, Lin Hu

**Affiliations:** 1School of Land and Resources Engineering, Kunming University of Science and Technology, Kunming 650093, China; biruikmust@163.com (R.B.); 18469110089@163.com (R.L.); kmust_gs@126.com (S.G.); gtzygc311@163.com (W.L.); hl112365@163.com (L.H.); 2Plication Engineering Research Center of Spatial Information Surveying and Mapping Technology in Plateau and Mountainous Areas Set by Universities in Yunnan Province, Kunming 650093, China; gskmust@163.com; 3College of Geosciences and Engineering, West Yunnan University of Applied Sciences, Dali 671000, China

**Keywords:** UAV, DJI Phantom 4 RTK, pit-rim landforms, repeated observation, oblique imaging-assisted, DSM of Difference

## Abstract

Unmanned Aerial Vehicles (UAVs) are a novel technology for landform investigations, monitoring, as well as evolution analyses of long−term repeated observation. However, impacted by the sophisticated topographic environment, fluctuating terrain and incomplete field observations, significant differences have been found between 3D measurement accuracy and the Digital Surface Model (DSM). In this study, the DJI Phantom 4 RTK UAV was adopted to capture images of complex pit-rim landforms with significant elevation undulations. A repeated observation data acquisition scheme was proposed for a small amount of oblique-view imaging, while an ortho-view observation was conducted. Subsequently, the 3D scenes and DSMs were formed by employing Structure from Motion (SfM) and Multi-View Stereo (MVS) algorithms. Moreover, a comparison and 3D measurement accuracy analysis were conducted based on the internal and external precision by exploiting checkpoint and DSM of Difference (DoD) error analysis methods. As indicated by the results, the 3D scene plane for two imaging types could reach an accuracy of centimeters, whereas the elevation accuracy of the orthophoto dataset alone could only reach the decimeters (0.3049 m). However, only 6.30% of the total image number of oblique images was required to improve the elevation accuracy by one order of magnitude (0.0942 m). (2) An insignificant variation in internal accuracy was reported in oblique imaging-assisted datasets. In particular, SfM-MVS technology exhibited high reproducibility for repeated observations. By changing the number and position of oblique images, the external precision was able to increase effectively, the elevation error distribution was improved to become more concentrated and stable. Accordingly, a repeated observation method only including a few oblique images has been proposed and demonstrated in this study, which could optimize the elevation and improve the accuracy. The research results could provide practical and effective technology reference strategies for geomorphological surveys and repeated observation analyses in sophisticated mountain environments.

## 1. Introduction

As UAV technology makes strides and is increasingly developed, UAVs have become progressively popularized and civilianized. In addition, UAV Remote Sensing (UAV-RS) is being gradually achieved by employing UAVs as carriers with imaging or non-imaging sensors to capture high-resolution remote sensing images, SAR images, as well as high-precision laser point clouds [1,2,3]. Since low-altitude and small UAVs exhibit several advantages (e.g., mobility, flexibility, low-cost, and rich data results) [4], they have been adopted to generate 3D scenes by integrating Structure from Motion (SfM) and Multi-View Stereo (MVS) algorithms [5] for landslide disaster monitoring [6], flood hazard assessment [7], geomorphological evolution analysis [8], as well as crop monitoring [9].

Geomorphology refers to various undulating forms in the earth’s surface, created under constant shaping by internal and external geological forces. It is one of the critical factors to explain and analyze the physical geographic environment [10]. UAVs have been adopted extensively to capture images of different landforms in different periods. The constructed 3D scenes, Digital Elevation Models (DEMs), and Digital Surface Models (DSMs) are applied for rapid, non-contact geomorphological surveys, geomorphological feature extraction and change analysis [11,12,13,14]. However, the accuracy of 3D-scene measurement, elevation error and the error of the instrument are recognized as the basis for ensuring qualitative and quantitative analyses of the geomorphological survey and its morphological evolution under a long time series and repeated observations. As reported by existing studies, 3D scene construction achieved by exploiting images acquired by UAVs equipped with Global Navigation Satellite System (GNSS) with low positioning accuracy fails to meet 3D measurement accuracy requirements [15], and the achieved data results are of less significance to multi-phase geomorphological evolution analysis studies [16]. To increase 3D-scene accuracy, Ground Control Points (GCPs) have been primarily applied as a constraint to improve the exterior orientation elements of the original image in aerial triangulation and to achieve absolute orientation [17,18]. Moreover, a data processing method that integrates GCPs and UAV images is capable of monitoring centimeter-scale variations of collapse erosion landform [19], open-pit mine dump erosion zone [20], as well as wetland tidal flats [21]. Likewise, the measurement accuracy of relevant glacial landforms [22], outcrop geology [23], and other studies can be met. However, the mountainous environment of the Yunnan plateau is complicated, and different landscape types are located in a wide variety of environments. Thus, GCPs are difficult to lay. An unreasonable GCPs layout will reduce the accuracy of the 3D scene, rendering repeat observations impossible. For complex terrain environments, UAVs equipped with a Real-Time Kinematic (RTK) system can capture images, and exploit differential correction data provided by Continuously Operating Reference Stations (CORS) to acquire high-precision positioning information [24] and achieve aerial triangulation without GCPs [25,26]. On that basis, 3D scene accuracy is improved to a greater extent, and operational risks and the dependence on GCPs are reduced [27]. Most existing studies for 3D scene accuracy are based on combining different data processing software and different GCPs layout schemes to accurately assess a single image dataset by using checkpoints [28,29]. However, when RTK UAVs are adopted to analyze the geomorphological evolution of long time series, the accuracy of the 3D scenes of repeated observations may change significantly. Furthermore, the repeated observation accuracy of RTK UAV has been rarely investigated, and its error analysis under no GCPs constraint has been scarce on the complex mountain environment terrain.

Thus, under no GCPs constraint, for the same time as without a topographic variation of complex mountain pit-rim landforms, DJI Phantom 4 RTK UAV was employed in this study to repeat the observation with ortho-view at a fixed altitude and overlap while adding repeated observation with a small number of oblique-view images by flight a section. The 3D scene construction was achieved by applying SfM-MVS technology for different datasets, obtained from repeated observations. By using checkpoints and DoD, the accuracy analysis was conducted by comparing different datasets with each other for internal and external precision.

## 2. Study Area

The study area is a typical circular pit-rim landscape at the southern edge of Dinosaur Valley on the west side of Lufeng Jurassic Dinosaur Site Park in Lufeng, Chuxiong, Yunnan (Figure 1a). The width is nearly 3 km from east to west, 10 km length from north to south, and 4 km in diameter. It is surrounded by steep mountains that connect to form a bar-shaped radial ridge in contrast to the central terrain. The complex pit-rim at the highest entrance in the northeast is the subject of the investigation (Figure 1b). The location of this area has a significant downward gradient from the external structure, and significant differences are identified in topography and features between the interior and exterior of the ridge (Figure 1c). Thus, the study area could be a promising test area for geomorphological data collection and repeated observation accuracy analyses.

## 3. Material and Methods

### 3.1. Data Acquisition

#### 3.1.1. UAV Images Collection

The DJI Phantom 4 RTK (P4R) UAV was equipped with a multi-frequency and multi-system high-precision RTK GNSS and an FC6310R camera (20-megapixel RGB sensor). It was also equipped with an active stabilizing camera cradle head, which ensures sharp images, to compensate for the unstable UAV flight conditions. Moreover, DJI GS RTK ground station control software was applied for route planning and image-data acquisition in the test area. Table 1 lists the specifications of the aircraft and camera.

The test area covered an area of 0.06 km^2^. The aerial survey campaign was conducted on 21 July 2021. The UAV was set at a 200 m flight altitude directly above the take-off point. To avoid missing image data and long flight time, the main route angle was set to 68°, and the direction of the route was set as perpendicular to the edge of the pit-rim. The forward and the side overlap were 80%. Moreover, the RTK status was fixed to obtain the differential signal returned from the CORS station to the UAV instantly. The traffic conditions, UAV take-off and landing points were determined by the Google Earth platform and through field research.

First, the lens inclination angle was set to −90° to collect orthophotos. Subsequently, it was set to −45°, and the UAV was flown along a straight line connecting the last waypoint to the center of the route-planning area to acquire partially oblique images (Figure 3a). The lens direction was kept parallel to the straight-line direction. Several camera parameters (e.g., ISO, exposure value and shutter) were automatically regulated by complying with the real brightness of the scene at the time of flight. The UAV flight parameters were selected as shown in Table 2.

Three flights were conducted based on the identical flight parameters, two of which were repeated observations, i.e., R1,R2,R3. Six datasets were developed by combining the data types “orthophotos” (O1) or “oblique imaging-assisted” (O2) (Table 3). To be specific, to distinguish different datasets, repeated observation numbers and image data types were adopted to name the respective dataset, e.g., 80%_R1_O1 represents the orthophoto dataset of the first observation, 80%_R2_O2  denotes the orthophotos and oblique imaging-assisted dataset of the second observation. Using the 80%_R1_O1 image dataset as an example, some orthophotos and oblique images are presented in Figure 2. The UAV type is shown in Figure 3c.

#### 3.1.2. Checkpoint Acquisition

Influenced by the significant difference between the inside and outside of the pit-rim landforms, the obvious feature points were adopted as checkpoints to analyze the measurement accuracy comprehensively. Overall, 27 checkpoints (Figure 3b) were acquired with the Hi-Target V90 RTK device (Figure 3d) by connecting to the CORS network, and the coordinate system used was CGCS 2000. This RTK equipment achieved a planimetric accuracy of ±8 mm + 1 ppm and an elevation measurement accuracy of ±15 mm + 1 ppm. 

### 3.2. 3D Scene Construction Based on SfM-MVS

The 3D scene construction was achieved using SfM-MVS technology [30] in Agisoft Metashape Professional 1.6.4 software. Under an identical data processing flow (Figure 4), six datasets were processed (Table 3) to generate 3D scenes, DOMs, as well as a 0.1 m resolution DSMs.

To begin with, in the SfM algorithm, the Scale Invariant Feature Transform (SIFT) algorithm [31] was employed to extract the feature points on each image. The scale, position, direction eigenvalues of the key points on the respective image were then determined. Subsequently, in accordance with the key points on the overlapping images, image feature matching was performed by combining the image attitude and spatial position information, the camera focal length, as well as radial and tangential distortion model parameters. Lastly, Bundle Block Adjustment (BBA) was performed [32]. The key point features and camera parameter positions were optimized by minimizing the reprojection error between the locations of key points on the image and the predicted locations [33], as an attempt to generate a sparse point cloud with coordinate and color information.

The MVS algorithm first iteratively diffused and filtered the sparse point cloud continuously to generate the dense point cloud and construct the 3D point cloud data [34]. Second, the dense point cloud was segmented into blocks, and the block point cloud was constructed on an irregular three-dimensional grid. Subsequently, the white film data were obtained from the grid data, and the optimal texture was automatically detected through the triangular network to automatically optimize texture mapping. Lastly, the 3D scene consistent with the dense point cloud coordinate system was obtained.

### 3.3. Accuracy Assessment Methods

#### 3.3.1. 3D scene Absolute Accuracy Assessment

To analyze the absolute accuracy of 3D scenes within a range of datasets, the deviation between the measured value of the model and the checkpoint in 3D scenes was determined. The mean error ($E_{mean}$) and Root Mean Square Error ($E_{RMSE}$) were adopted to quantify the plane accuracy and elevation accuracy. The calculation formula as:(1)$$E_{mean}=\frac{\sum_{i=1}^{n}\left(X_{i}^{\prime }-X_{i}\right)}{n}E_{RMSE}=\sqrt{\frac{\sum_{i=1}^{n}{\left(X_{i}^{\prime }-X_{i}\right)}^{2}}{n}}$$
where n denotes the number of checkpoints; $X_{i}^{\prime }$ represents the image solution value for the respective direction of checkpoints; $X_{i}$ denotes the measured value of the respective direction of checkpoints; $E_{RMSE_{x}}$, $E_{RMSE_{y}}$, $E_{RMSE_{z}}$ are the RMSE of each direction of the checkpoint.

#### 3.3.2. Internal and External Precision Assessment

The accuracy of the terrain 3D scene constructed by SfM-MVS technology was analyzed in terms of internal and external precision [30], which revealed the differences that led to the sources of errors in 3D scenes and the mechanisms of their transmission features. In this study, the internal precision was the error generated by the repeated observation data in the SfM-MVS technology, while the external precision was the error caused by the change of the number and position of the oblique images.

To be specific, the internal precision between pairs of repeated observations was compared, while the dataset type remained unchanged, as follows: 80%_R1_O1 and 80%_R2_O1, 80%_R1_O1 and 80%_R3_O1, 80%_R2_O1 and 80%_R3_O1 for the orthophoto datasets, 80%_R1_O2 and 80%_R2_O2, 80%_R1_O2 and 80%_R3_O2, 80%_R2_O2 and 80%_R3_O2, for the oblique imaging-assisted datasets. The external precision complied with the 80%_R2_O2 dataset to compare the numbers and location schemes of four different oblique imaging datasets (Figure 5): (1) two images in the end or middle of the oblique flight route, i.e., $80\%\_R2\_2Photos\;and\;80\%\_R2\_2Photos\left(Mid\right)$. (2) Four images were selected at equal intervals, i.e., 80%_R2_4Photos. (3) Six consecutive images were selected in the middle, i.e., 80%_R2_6Photos.

Furthermore, before the quantitative analysis of the internal and external precision of the repeated observation datasets, the different values of the checkpoint model values in different 3D scenes should be compared, and the $E_{mean}\;\mathrm{and}\;E_{RMSE}$ for the respective axial, plane, and the 3D should be calculated. Moreover, the DoD was able to visualize the variations of surface morphology between DSMs in different periods and analyze the elevation deviation [35]. Accordingly, DoD differences were processed in ArcMap, and the histogram statistics were performed in accordance with the calculated results.

## 4. Results and Analysis

### 4.1. 3D Scene Absolute Accuracy Analysis

The 3D scene accuracy assessment results for different datasets were analyzed using checkpoints (Table 4), which indicated that the magnitude of the plane error of the six datasets was consistent, whereas the magnitude of the elevation error was significantly different. Under the different image data types, the repeated observation error results were constant.

For the three repeated observations of orthophoto datasets (O1), the average plane error was 0.0445 m, the average 3D error was 0.3082 m, and the elevation error was 0.3049 m. For oblique imaging-assisted datasets (O2), the average plane error was 0.0339 m, the average 3D error was 0.1005 m, the elevation error reached 0.0942 m, and the plane and elevation errors were found to be within centimeters. In the orthophoto datasets (O1), the elevation accuracy could only reach the decimeters. However, by adding the oblique images (O2), in the three repeated observations, only the last (R3) elevation error reached the decimeters, whereas the others (R1, R2) were within centimeters. The overall elevation accuracy was improved by one order of magnitude. 

Both types of imaging datasets could achieve the plane accuracy of a centimeter order of magnitude for complex mountainous terrain. Increasing the number of oblique images could improve the overall elevation accuracy, reduce the fieldwork required, and significantly improve work efficiency and safety.

### 4.2. Internal Precision Analysis

To analyze the effect of UAV and SfM-MVS technology on the internal accuracy of repeated observations, 3D scene checkpoint model values and DoD were analyzed for repeated observations under the identical data type (Table 5).

Compared with the repeated observation results under the orthophoto datasets (O1), which achieved the average plane error of 0.0405 m and the average 3D error of 0.1035 m, the oblique imaging-assisted datasets (O2) had a mean plane error of 0.0417 m and an average 3D error of 0.0693 m, for which the elevation deviation of repeated observations was smaller.

Impacted by the obvious differences in elevation between different datasets, the elevation error was able to fully reflect the stability of UAV and SfM-MVS technology for repeated observations. The correlation between repeated observations of orthophoto datasets (O1) (Figure 5a) exceeded 0.84, the correlation between oblique imaging-assisted datasets (O2) (Figure 6b) exceeded 0.80, and the correlation between 80%_R1_O1 and 80%_R1_O2 was 0.92, which revealed that the elevation error between repeated observations of P4R UAV for the 3D reconstruction of terrain based on SfM-MVS technology was stable.

In addition, according to the DoD results of different datasets under repeated observations (Figure 7, Table 6), the mean absolute value of error between DSMs of orthophoto datasets (O1) was 0.13 m and the mean standard deviation reached 0.2024 (Figure 7a–c). For oblique imaging-assisted datasets (O2), the mean absolute value was 0.07 m, and the mean standard deviation was 0.1914 (Figure 7d–f). Combined with the standard deviation and normal distribution curves (Figure 7), no significant difference was reported in the dispersion of the internal precision error of repeated observations between the two imaging data types. However, assisting the oblique images could reduce the error, and the internal precision of repeated observation was stable.

### 4.3. External Precision Analysis

Before analyzing the effect of oblique imaging-assisted datasets on external precision, the checkpoints error of 3D scenes with 4 different oblique imaging numbers and position schemes were analyzed (Table 7). The plane error of the 4 schemes was better than 0.04 m, reaching a centimeter order of magnitude, and the elevation error was better than 0.14 m. Compared with the results of orthophoto datasets (O1), the elevation error was reduced by 2.28 times (0.3049 m to 0.1338 m) by adding 2 oblique images. A few oblique images could effectively improve the elevation precision. Compared with the results of 2 oblique images, the elevation error increases to 0.1385 m by adding 4 oblique images, and the method of equal interval extraction of oblique images could not effectively reduce the elevation error.

Using 80%_R2_O2 DSM as a reference, the DoD results of different schemes (Figure 8, Table 8) revealed that the mean absolute values of elevation deviations between different datasets ranged from 0.0415 m to 0.0772 m, the standard deviations ranged from 0.1208 to 0.1518, and the number of oblique images gradually decreased from 6 to 2. When 2 oblique images were added, the average absolute value was 0.0772 m. When the positions of the 2 images were changed, the average absolute value was 0.0489 m, and the absolute value difference was insignificant, whereas the standard deviation increased to 0.1518, and the error distribution was relatively discrete. When 4 oblique images were added, the average absolute value was 0.0865 m. The scheme exerted the worst effect on elevation optimization. The average absolute values of 6 and 2 oblique images were similar, and the elevation error was the smallest. For the 3D reconstruction of a complex mountain environment, the elevation accuracy could be optimized by adding 2 oblique images located in the middle of the oblique route, which could also meet the precision requirements of a correlation analysis.

## 5. Discussion

At present, greater focus has been placed on the measurement accuracy analysis of different types of UAVs under the constraints of GCPs in various fields. The measurement accuracy is largely dependent of GCPs, whereas the possibility of RTK UAV measurement without using GCPs is commonly overlooked. Thus, this study employed a repeated observation method assisting a small number of oblique-view images while observing the image from the ortho-view. In addition, the measurement accuracy of the RTK UAV without GCPs was analyzed. To conduct a targeted analysis, a typical pit-rim landform with significant differences in undulation at the southern edge of Lufeng Dinosaur Valley was adopted as the test area for a comparative test study. As revealed by the elevation deviation and histogram statistical results, which were obtained by checkpoint error and DoD, the accuracy optimization feasibility of repeated observation 3D measurement results of various data was obtained, displaying internal and external precision.

As indicated by the checkpoint error analysis, the plane error of the 3D scene under the two types of datasets could reach centimeters, RTK UAV could effectively improve the plane accuracy, and the effect was significant for sophisticated mountainous terrain environments. However, the elevation values of the two imaging types were significantly different, and the results of the oblique imaging-assisted datasets (O2) were better than those of the orthophoto datasets (O1). Moreover, there was a vertical offset between the DSMs, obtained by repeated observations, which partially attributed to the change of the estimated focal length (f) in self-calibration. Table 9 lists the statistical results of the focal length estimates for different datasets.

Table 9 lists the same camera focal length estimates (3707.94 pix) obtained from repeated observations of the orthophoto datasets (O1), and the oblique imaging-assisted datasets (O2), with an average value of 3708.14 pix. According to the checkpoint error results, the addition of oblique images could be used across a larger range of perspectives, and effectively regulate the estimated camera focal length in the solving process. The estimation results of the oblique imaging-assisted datasets (O2) were found to be more accurate.

Though the elevation deviation of the two image data types is significant, the elevation error correlation is found to be more intense between the repeated observation datasets of the respective image data type, where the internal accuracy is stable, and SfM-MVS technology exhibits a high reproducibility for the 3D scene construction of terrain and its derived results. UAVs are capable of capturing high-resolution images with a considerable amount of image information, and complete details of ground objects. Moreover, the datasets obtained using SfM-MVS technology have low network roughness and provide an optimized data perspective, which effectively reduces the size of the shadow [36]. Accordingly, SfM-MVS technology underpins an effective and quantitative change in the monitoring of different landforms, while obtaining high-precision and high-quality results. To be specific, Alfredsen et al. [37] successfully achieved feature-element extraction and an accurate calculation of river ice. Pagan et al. [38] analyzed the historical evolution of dunes and their relationship with coastal erosion. Yang et al. [39] analyzed the landslide evolution by obtaining multi-period landslide images and extracting feature elements of landslides in different periods.

By analyzing the external precision of different oblique imaging numbers and position change schemes, the elevation error tends to decrease, and the error distribution is found to be more concentrated and stable with an increase in the oblique imaging number. To further analyze the effect of the image position on the external accuracy, the distance between 2 images was determined by complying with the Position and Orientation System (POS) information, and the mean distance of the respective scheme was obtained. As impacted by the change of image position, it also represents the change of overlap on the oblique route. The overlap between images was calculated through the feature matching method [40]. The results are listed in Table 10.

Table 10 lists the mean image spacing and overlap results under different oblique imaging numbers and position schemes. For the improved scheme of two middle and six oblique images, the image spacing was less than 40 m, and the image overlap exceeded 80%. For the two image schemes at both ends, under the significant image spacing, there was found to be no certainty overlap. For the four images extracted with equal intervals, the image spacing was doubled, and the overlap was reduced to less than 67%. Furthermore, the image spacing impacts the elevation error. If the image overlap is not lower than the preset parameter, the image distribution will be more concentrated, and the error reduced. For the mentioned reasons, the elevation accuracy from the data acquisition method will be optimized, and the overall 3D measurement accuracy will be effectively improved.

## 6. Conclusions

In this study, DJI Phantom 4 RTK UAV was employed to collect repeated observation images of complex mountain pit-rim landforms via ortho-view, while adding a small amount of repeated observation from oblique-view images. For various imaging datasets, SfM-MVS technology was adopted to realize 3D scene construction. The comparative analysis methods of checkpoint and DoD were employed to compare different datasets as well as the 3D measurement accuracy of repeated redundant observation for the internal and external precision. As indicated by the results of the checkpoint comparison, with other parameters unchanged, regardless of whether it is a single orthophoto observation or considering the 3D scene formed by oblique images, the plane error reaches centimeters, whereas the elevation error is found to be significantly different. The average elevation error of the 3D scene constructed by orthophoto datasets was 0.3049 m, while the average elevation error of the oblique imaging datasets was only 0.0942 m, thereby the elevation accuracy was significantly improved, and the number of oblique images accounted for only 6.3% of the total number of images. This means that the proposed method of performing RTK UAV orthophotography, while appropriately assisting oblique imaging acquisition technology can effectively improve elevation accuracy in constructing 3D scenes of complex surface environments. This method is capable of reducing the dependence on GCPs and the field workload. Moreover, the empirical study indicates that the terrain 3D reconstruction by using SfM-MVS technology exhibited effective reproduction. The repeated observation elevation deviation of the oblique imaging was smaller, and the internal precision was stable. When the number or location of oblique images were changed, the external precision was effectively improved. The elevation error and standard deviation were gradually reduced, and the error distribution was more concentrated and stable. It is clear that with the smaller distance between oblique images, the elevation accuracy can be further optimized.

This study primarily analyzed repeated observations under constant flight altitude, the same overlap, as well as oblique-imaging data. However, UAVs are affected by the natural environment when collecting images, which causes blurred images and inconsistent color and thus affects the extraction of image feature points to a certain extent. The effects of different flight altitudes, overlap and different oblique imaging collection methods on the repeated observation accuracy require further analysis.

## Figures and Tables

**Figure 1 sensors-21-08109-f001:**
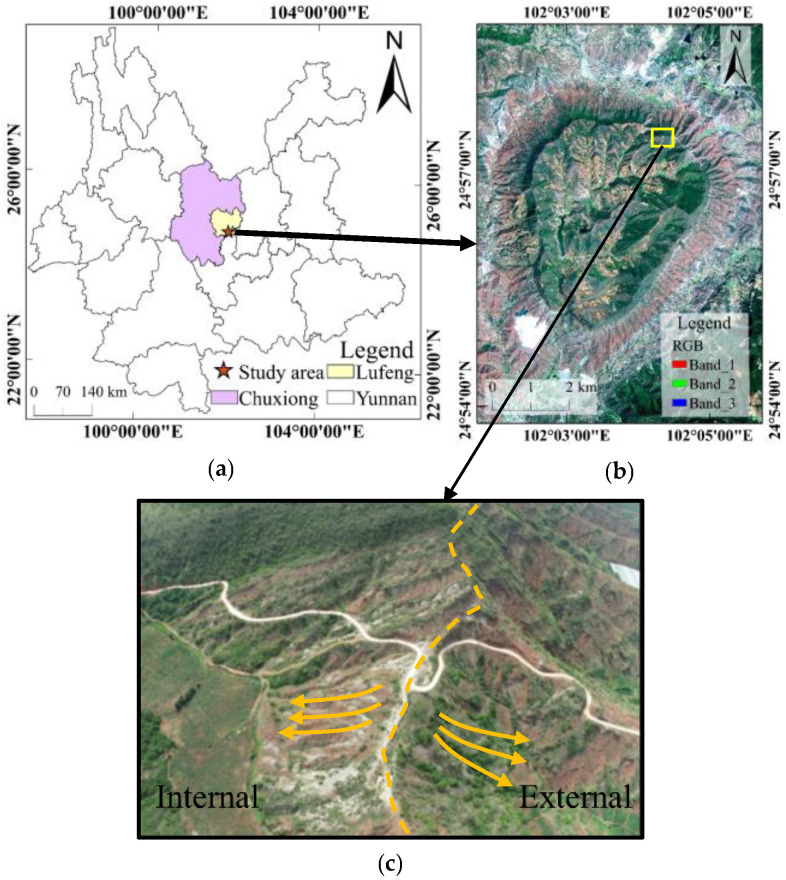
(**a**) Location of Lufeng; (**b**) Satellite image of the study area; (**c**) Complex pit-rim at the entrance.

**Figure 2 sensors-21-08109-f002:**
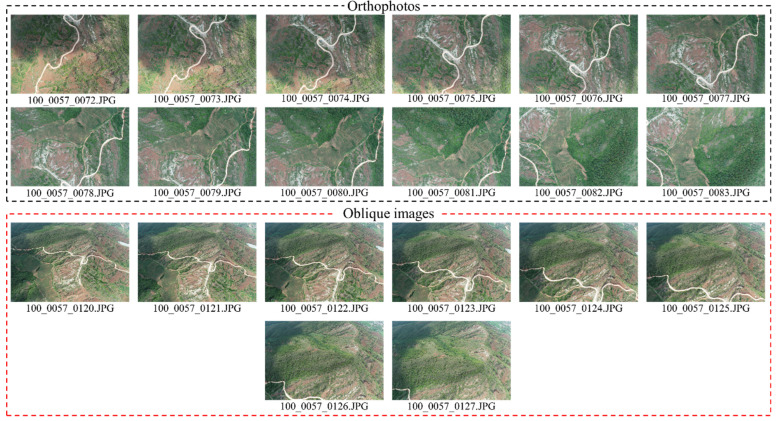
Partial orthophotos and oblique images (80%_R1_O1  dataset as an example).

**Figure 3 sensors-21-08109-f003:**
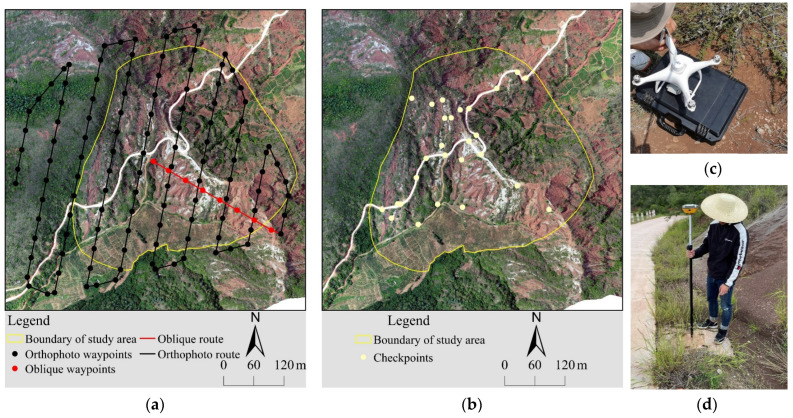
(**a**) UAV flight routes; (**b**) Location of checkpoints; (**c**) UAV type; (**d**) RTK data acquisition.

**Figure 4 sensors-21-08109-f004:**
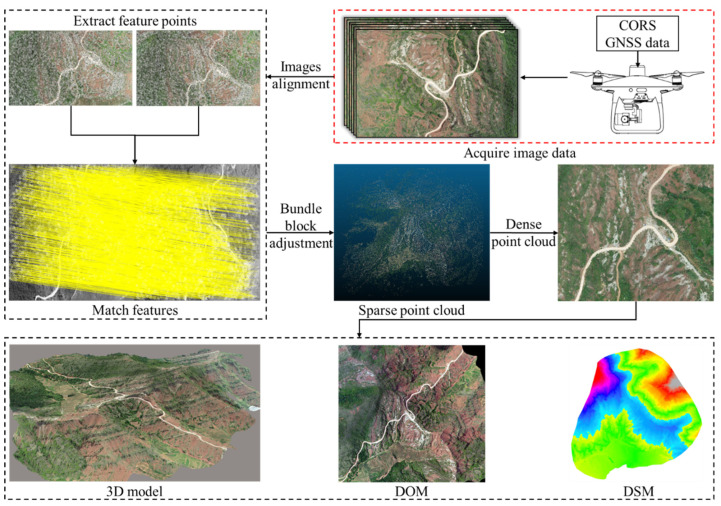
3D scene construction process of SfM-MVS technology.

**Figure 5 sensors-21-08109-f005:**
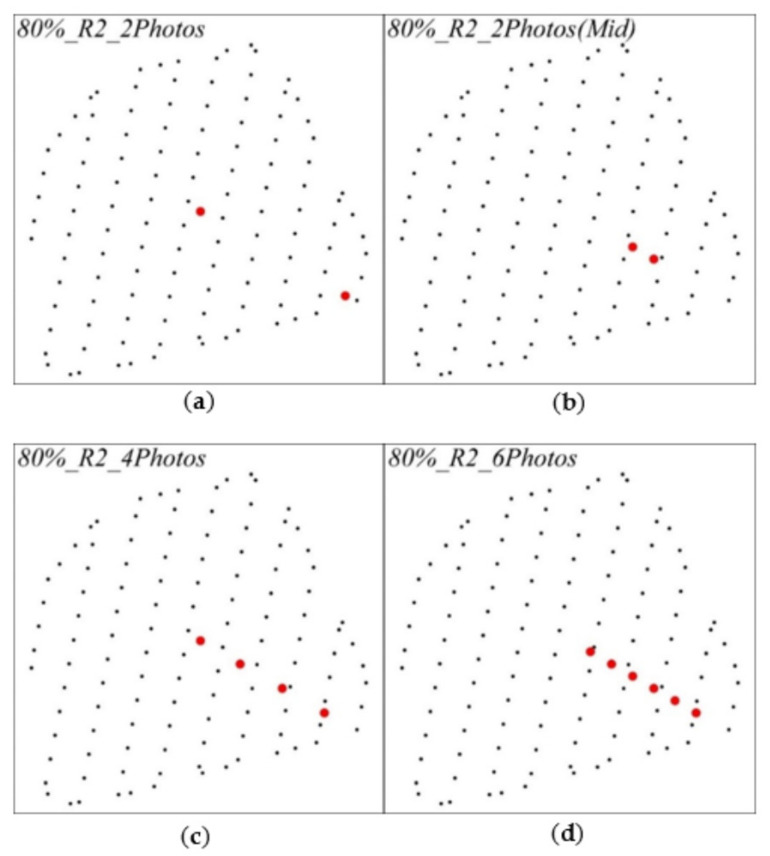
4 different oblique imaging numbers and location schemes, (**a**) 2 images in the end of the oblique flight route, (**b**) 2 images in the middle of the oblique flight route, (**c**) 4 images at equal intervals of the oblique flight route, (**d**) 6 consecutive images in the middle of the oblique flight route. (Black points are orthophoto waypoints, red points are oblique imaging waypoints).

**Figure 6 sensors-21-08109-f006:**
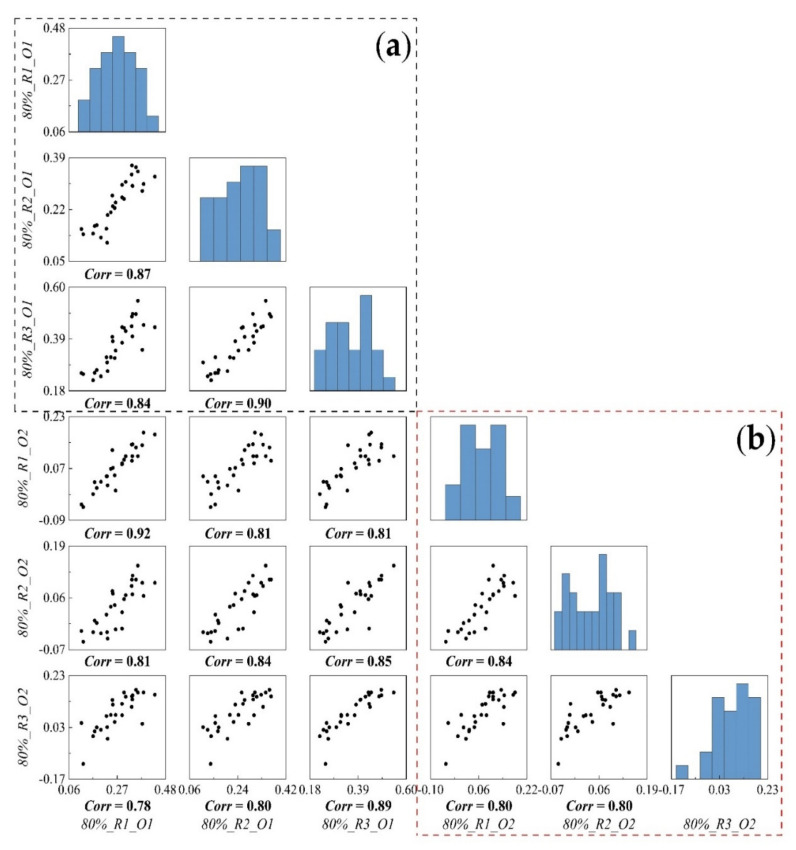
The elevation error correlation results of different datasets under repeated observations, *Corr* is the correlation coefficient, (**a**) represents the orthophoto datasets, (**b**) represents the oblique imaging datasets.

**Figure 7 sensors-21-08109-f007:**
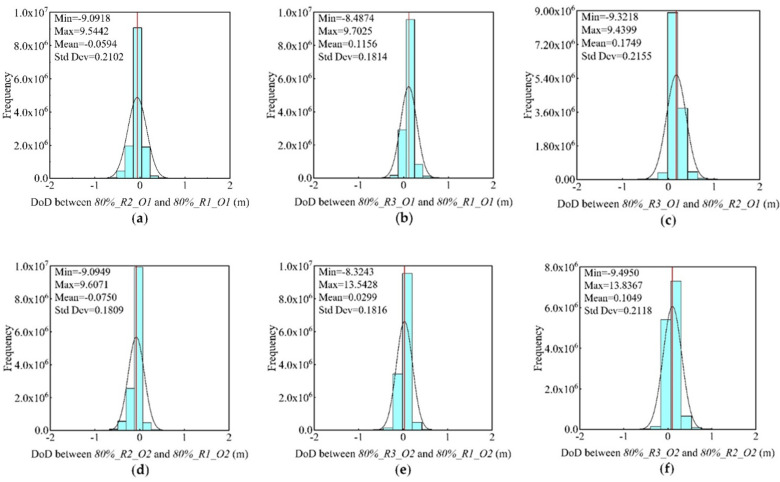
DoD results of different datasets under repeated observations. Histograms (**a**–**c**) represent the orthophoto datasets results. Histograms (**d**–**f**) represent the oblique imaging datasets.

**Figure 8 sensors-21-08109-f008:**
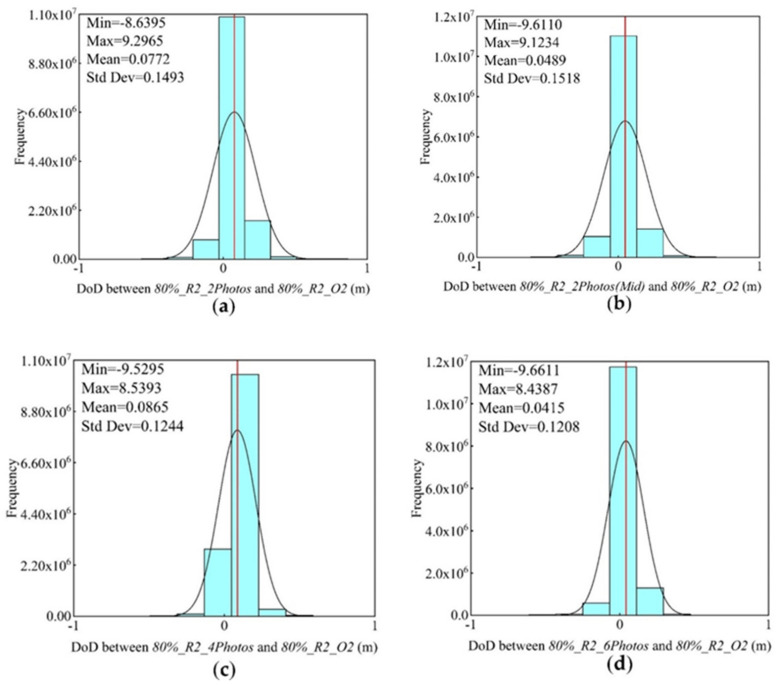
DoD results of different oblique imaging numbers and location schemes. Histogram (**a**) represents the result of 2 images in the end of the oblique flight route, histogram (**b**) represents the result of 2 images in the middle of the oblique flight route, histogram (**c**) represents the result of 4 images at equal intervals of the oblique flight route, histogram (**d**) represents the result of 6 consecutive images in the middle of the oblique flight route.

**Table 1 sensors-21-08109-t001:** Aircraft and camera specifications.

UVA Specifications			Camera Specifications	
Aircraft		DJI Phantom 4 RTK	Sensor	FC6310R
Take-off weight		1391 g	Sensor format	1” (CMOS) 13.2 mm × 8.8 mm
Max flight speed		50 km/h (Positioning) 58 km/h (Attitude)	Focal length	8.8 mm
Max flight time		30 min	35 mm equiv. focal length	24 mm
Max take-off altitude		6000 m	Image resolution	5472 × 3648
Satellite positioning systems	Asia	GPS + BeiDou + Galileo	Field of View	84°
	Other area	GPS + GLONASS + Galileo		
GNSS positioning accuracy	Horizontal	1 cm + 1 ppm (RMS)	Pixel size	2.41 µm
	Vertical	1.5 cm + 1 ppm (RMS)		

**Table 2 sensors-21-08109-t002:** UAV flight parameters.

UAV Flight Parameters		
Type of Image	Orthophoto	Oblique Image
Survey date	July 2021	
Flight altitude (m)	200 m	
Ground Sampling Distance (GSD, cm/px)	5.48 cm/px	
Image forward overlap (%)	80%	
Image side overlap (%)	80%	
Main route angle (°)	68°	
Camera angle (°)	−90°	−45°
RTK status	Fixed	
Flight time (min, s)	10 min	35 s
Number of images captured	119	8

**Table 3 sensors-21-08109-t003:** Characteristics of datasets resulting from the test setup.

Dataset	Type of Image	Number of Images	Area (km^2^)	Number of Tie-Points	Mean Density of Tie-Points (pts/m^2^)
80%_R1_O1	orthophoto	119	0.0647	138,924	0.35
80%_R2_O1			0.0619	123,707	0.30
80%_R3_O1			0.0648	136,996	0.35
80%_R1_O2	Oblique image	127 (8 of oblique images)	0.0660	144,410	0.36
80%_R2_O2			0.0671	128,634	0.30
80%_R3_O2			0.0666	144,407	0.36

**Table 4 sensors-21-08109-t004:** 3D scenes accuracy assessment results for different datasets.

Dataset	Error (m)				
	$E_{RMSE_{x}}$	$E_{RMSE_{y}}$	$E_{RMSE_{z}}$	$E_{RMSE_{xy}}$	$E_{RMSE_{xyz}}$
80%_R1_O1	0.0265	0.0338	0.2832	0.0430	0.2865
80%_R2_O1	0.0232	0.0380	0.2554	0.0445	0.2593
80%_R3_O1	0.0311	0.0340	0.3761	0.0461	0.3789
$E_{Mean}$	0.0269	0.0353	0.3049	0.0445	0.3082
80%_R1_O2	0.0240	0.0231	0.0968	0.0333	0.1024
80%_R2_O2	0.0191	0.0300	0.0683	0.0356	0.0770
80%_R3_O2	0.0182	0.0272	0.1175	0.0327	0.1220
$E_{Mean}$	0.0204	0.0267	0.0942	0.0339	0.1005

**Table 5 sensors-21-08109-t005:** The difference results of 3D scene checkpoint model values of different datasets under the identical data type.

Dataset 1	Dataset 2	Error (m)				
		$E_{RMSE_{x}}$	$E_{RMSE_{y}}$	$E_{RMSE_{z}}$	$E_{RMSE_{xy}}$	$E_{RMSE_{xyz}}$
80%_R2_O1	80%_R1_O1	0.0300	0.0289	0.0487	0.0416	0.0641
80%_R3_O1	80%_R1_O1	0.0281	0.0272	0.1058	0.0391	0.1128
80%_R3_O1	80%_R2_O1	0.0288	0.0290	0.1273	0.0409	0.1337
$E_{Mean}$		0.0290	0.0284	0.0939	0.0405	0.1035
80%_R2_O2	80%_R1_O2	0.0304	0.0301	0.0477	0.0427	0.0640
80%_R3_O2	80%_R1_O2	0.0240	0.0302	0.0521	0.0386	0.0648
80%_R3_O2	80%_R2_O2	0.0276	0.0339	0.0657	0.0437	0.0790
$E_{Mean}$		0.0273	0.0314	0.0552	0.0417	0.0693

**Table 6 sensors-21-08109-t006:** Statistics of DoD internal precision results of different datasets under repeated observations.

Dataset 1	Dataset 2	Error (m)			Std Dev
		Minimum	Maximum	Mean	
80%_R2_O1	80%_R1_O1	−9.0918	9.5442	−0.0594	0.2102
80%_R3_O1	80%_R1_O1	−8.4874	9.7025	0.1156	0.1814
80%_R3_O1	80%_R2_O1	−9.3218	9.4399	0.1749	0.2155
$\left|Mean\right|$		8.9670	9.5622	0.1166	0.2024
80%_R2_O2	80%_R1_O2	−9.0949	9.6071	−0.0750	0.1809
80%_R3_O2	80%_R1_O2	−8.3243	13.5428	0.0299	0.1816
80%_R3_O2	80%_R2_O2	−9.4950	13.8367	0.1049	0.2118
$\left|Mean\right|$		8.9514	12.3289	0.0700	0.1914

**Table 7 sensors-21-08109-t007:** 3D scenes accuracy assessment results of 4 different oblique imaging numbers and position schemes.

Dataset	Error (m)				
	$E_{RMSE_{x}}$	$E_{RMSE_{y}}$	$E_{RMSE_{z}}$	$E_{RMSE_{xy}}$	$E_{RMSE_{xyz}}$
80%_R2_2Photos	0.0264	0.0244	0.1338	0.0360	0.1385
$80\%\_R2\_2Photos\left(Mid\right)$	0.0250	0.0225	0.1234	0.0336	0.1279
80%_R2_4Photos	0.0270	0.0223	0.1385	0.0350	0.1429
80%_R2_6Photos	0.0225	0.0205	0.1225	0.0305	0.1262
80%_R2_O2	0.0191	0.0300	0.0683	0.0356	0.0770

**Table 8 sensors-21-08109-t008:** Statistics of DoD external precision results of different oblique imaging numbers and location schemes.

Dataset 1	Dataset 2	Error (m)			Std Dev
		Minimum	Maximum	Mean	
80%_R2_2Photos	80%_R2_O2	−8.6395	9.2965	0.0772	0.1493
$80\%\_R2\_2Photos\left(Mid\right)$		−9.6110	9.1234	0.0489	0.1518
80%_R2_4Photos		−9.5295	8.5393	0.0865	0.1244
80%_R2_6Photos		−9.6611	8.4387	0.0415	0.1208

**Table 9 sensors-21-08109-t009:** The results of focal length estimates for different datasets.

Dataset	f(Pix)	Mean
80%_R1_O1	3707.94	3707.94
80%_R2_O1	3707.94	
80%_R3_O1	3707.94	
80%_R1_O2	3707.94	3708.14
80%_R2_O2	3708.23	
80%_R3_O2	3708.24	

**Table 10 sensors-21-08109-t010:** The mean image spacing and overlap results under different oblique imaging numbers and position schemes.

Dataset	Mean Image Spacing (m)	Mean Image Overlap (%)
80%_R2_2Photos	268.6780	0
$80\%\_R2\_2Photos\left(Mid\right)$	39.2523	83.16
80%_R2_4Photos	78.8377	66.89
80%_R2_6Photos	39.4021	85.18
80%_R2_O2	38.3829	84.58

## Data Availability

The data are not publicly available as they involve the subsequent application of other studies.

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
