# Peer review of "Studies on Three-Dimensional (3D) Accuracy Optimization and Repeatability of UAV in Complex Pit-Rim Landforms As Assisted by Oblique Imaging and RTK Positioning"

_sensors, 2021, doi:10.3390/s21238109_

Round 1

Reviewer 1 Report

The authors have proposed a repeated observation method that only aids a few oblique images to participate in processing, which can achieve elevation optimization and accuracy improvement. The research results can provide practical and effective technology reference strategies for geomorphological surveys and repeated observation analysis in complex mountain environments.

Results seem to be promising. I recommend it for publication after major improvements.

Comments:

  1. The author needs to discuss in more detail the methods used, such as the parameters used, the devices, the running time, etc.

  1. There are some formatting errors, such as line breaks in lines 252 and 330.Please correct the errors.

  1. Some of the results of the experiments are wrong, 0.2677 in Table 2 should be 0.0267

  1. The authors should provide some images in the dataset.

Author Response

Dear reviewer, We are very grateful for your effort and time in reviewing our manuscript. We made modifications and improvements based on the comments. Please see the attachment, thank you. 

Reviewer 2 Report

Paper has merit, I didn't know the nadir vs oblique would have this kind of impact.

Clear problem statement, clear amd easy to understand  proof of evidence,  clear conclusions.

Author Response

Dear Reviewer,

We deeply appreciate the effort and time you’ve spent in reviewing our manuscript. We have made modifications and improvements according to the comments.

Please see the attachment, thanks.

Round 2

Reviewer 1 Report

This paper can be accepted in present form.